# Risk factors for fluoroquinolone- and macrolide-resistance among swine *Campylobacter coli* using multi-layered chain graphs

**Christine A. Wang**[1]**, William J. Love**[1]**, Manuel Jara**[1]**, Arnoud H. M. van Vliet**[2]**, Siddhartha Thakur**[1]**, Cristina Lanzas** [1]*

1 Department of Population Health and Pathobiology, College of Veterinary Medicine, North Carolina State University, Raleigh, North Carolina, United States of America, 2 School of Veterinary Medicine, Faculty of Health and Medical Sciences, University of Surrey, Guildford, United Kingdom

* clanzas@ncsu.edu

**Data availability statement:** All relevant data are within the manuscript and its Supporting information files.

## Abstract

*Campylobacter spp.* resistant to fluoroquinolones and macrolides are serious public health threats. Studies aiming to identify risk factors for drug-resistant *Campylobacter* have narrowly focused on antimicrobial use at the farm level. Using chain graphs, we quantified risk factors for fluoroquinolones- and macrolide-resistance in *Campylobacter coli* isolated from two distinctive swine production systems, conventional and antibiotic-free (ABF). The chain graphs were learned using genotypic and antimicrobial susceptibility data from 1082 isolates and host exposures obtained through surveys for 14 cohorts of pigs. The gyrA T86I point mutation alone explained at least 58% of the variance in ciprofloxacin minimum inhibitory concentration (MIC) for ABF and 79% in conventional farms. For macrolides, genotype and host exposures explained similar variance in azithromycin and erythromycin MIC. Among host exposures, heavy metal exposures were identified as risk factors in both conventional and ABF. Chain graph models can generate insights into the complex epidemiology of antimicrobial resistance by characterizing context-specific risk factors and facilitating causal discovery.

## Author summary

Antimicrobial resistance is influenced by multiple factors, including exposures to selecting agents, such as antibiotics, antiseptics, or heavy metals, and factors affecting the transmission of resistant pathogens, such as biosecurity and hygiene measures. Understanding what specific factors are associated with resistance in a given context is challenging. We developed an approach based on probabilistic graphical models to investigate context-specific antimicrobial resistance risk factors. We applied the approach to *Campylobacter coli* isolated from pigs in antibiotic-free and conventional farms.

**Funding:** This work was supported by the National Institutes of Health (R35GM134934 to CL and F30OD030022 to CAW, www.nih.gov). The funders had no rule in study design, data collection and analysis, decision to publish, or preparation of the manuscript.

**Competing interests:** The authors have declared that no competing interests exist.

We demonstrated how for fluoroquinolones, risk factors were similar across both types of farms, but risk factors for macrolides were different across settings.

## Introduction

*Campylobacter jejuni* and *C. coli* are common causes of human gastroenteritis [1]. In particular, drug-resistant *Campylobacter* is classified as a serious public health threat by the Centers for Disease Control and Prevention's (CDC) report on antibiotic resistant threats [2]. Every year in the United States alone, *Campylobacter* spp. causes an estimated 1.5 million infections, 29% of which have decreased susceptibility to fluoroquinolones or macrolides [2]. These antibiotic classes are used to treat severe *Campylobacter* infections in older and immunocompromised patients [3]. While *C. jejuni* is the most frequent source of infection, *C. coli* tends to have higher levels of drug resistance compared to *C. jejuni* [4–6]. *Campylobacter* is often acquired from consuming contaminated food and water and through direct contact with animals [1,7]. Food animals are reservoirs for both *Campylobacter* species, thus understanding the factors that drive *Campylobacter* antimicrobial resistance at the farm level is necessary to control and mitigate drug-resistant *Campylobacter* effectively [8].

Studies aiming to identify risk factors for drug-resistant *Campylobacter* have narrowly focused on antimicrobial use [9]. While antimicrobial use is a widely established risk factor for antimicrobial resistance among enteric pathogens, such as *Campylobacter* [10,11], antimicrobial use only captures a subset of all potential exposures that may drive resistance outcomes [11]. Heavy metals and biocides may also select for antimicrobial resistance [12]. Biosecurity and husbandry practices that modify microbial interactions can also influence resistance dynamics in agricultural settings [13]. Additionally, while antimicrobial exposure imposes selection pressures for resistant microorganisms, reduced or complete absence of antimicrobial exposure often does not correspond to either a reduction in bacterial resistance or reversion to susceptibility [14–16]. For example, resistant bacterial strains are found in organic and antibiotic-free farms [17], and resistance to banned antibiotics for food animals, such as chloramphenicol, persists in farms decades after the ban implementation [18]. An analysis of resistance drivers must include a comprehensive assessment of exposures beyond antimicrobial use.

Recent advances in data collection, including sequencing technologies, have made exposure and outcome data to assess risk factors for antimicrobial resistance more available and accessible than ever before [19]. However, the large number of exposures and outcomes presents challenges for classic analytic methods traditionally used in epidemiology. Regression-based methods are well suited for understanding relationships between a single disease outcome of interest and a relatively small number of potential exposures [20]. However, bacteria can be resistant to multiple antimicrobial drugs, and understanding the relationship between multiple resistant outcomes can yield additional information on their selection [21]. Additionally, ecological, evolutionary, and epidemiological processes at the microbial, host, and environmental levels can directly or indirectly influence the occurrence of resistance among microbial populations in a context-specific manner [11,17]. Therefore, the number of exposures we may want to consider is large and we may not have enough prior knowledge to select the relevant ones before the analysis.

Probabilistic graphical models are an effective framework for handling complex distributions in high dimensional space [22]. These statistical models are composed of random variables (represented as nodes) and edges encoding direct dependence relationships among

nodes [22]. They are classified based on the type of edges included in the models. Markov networks are a type of probabilistic graphical models with undirected edges, signifying that the directionality of the interaction between the connected variables cannot be ascribed [22]. Bayesian networks have only directed edges, indicating directional (causal) interactions. Chain graphs have both directed and undirected edges. Thus, they can represent a wider range of systems than Markov and Bayesian networks. Data from epidemiological studies of antimicrobial resistance often includes phenotypic and/or genotypic resistance and exposure variables. This data structure can be represented in multi-layer chain graphs (Fig 1). Antimicrobial susceptibility outcomes (minimum inhibitory concentrations (MIC)) can be represented as a layer with undirected edges with the nodes in the layer, as antimicrobial susceptibility outcomes can be correlated among themselves [21]. However, exposure and genotypic variables can explain part of the variation among these phenotypic outcomes. Thus, directed edges can link exposure layers (host exposures and/or microbial genotypes) to phenotypic outcomes. Recent advances in graph learning for mixed and multi-layer data are quickly expanding our ability to learn realistic graph representations from data [23–27]. Although the application of probabilistic graphical models in large epidemiological datasets has been scarce to date, their application as integration tools of antimicrobial resistance data is promising [28,29].

We aimed to identify risk factors for macrolide- and fluoroquinolone-resistance *Campylobacter coli (C. coli)* in two distinctive swine production systems: conventional and antibiotic-free (ABF). Previous research on *C. coli* populations in these systems found considerable resistance to fluoroquinolone and macrolide antibiotics [30]. Most notably, the prevalence of macrolide resistance was over 20% and not significantly different between bacterial isolates from conventional versus ABF production systems. Thus, we hypothesize that risk factors related to agricultural biosecurity practices and non-antimicrobial selection pressures would be significant predictors of macrolide resistance. On the other side, fluoroquinolone resistance was higher in conventional farms. Thus, we hypothesize that antimicrobial use is a main risk factor for fluoroquinolone resistance. To investigate the epidemiology of macrolide- and fluoroquinolone-resistant *C. coli*, we applied chain graph models and learned the model structure using genotypic resistance and phenotypic antimicrobial susceptibility data from *C. coli* isolates and host exposure data collected through surveys.

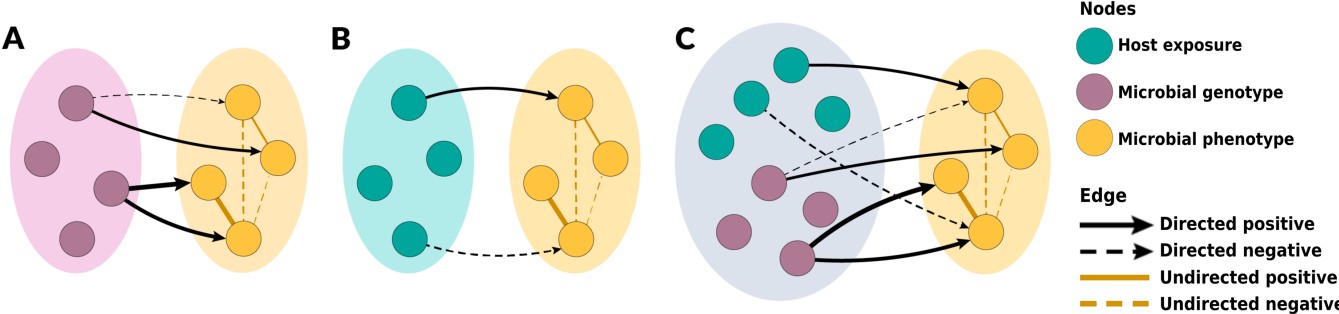

**Fig 1. Practical framework of specific chain graph analyses to be learned in this study.** Microbial antimicrobial susceptibility variables were used as the outcome variables for all chain graphs in this study. The directed edges between layers are described by $\beta$ and the undirected edges connected nodes within the outcome layer are described by $\Omega$. **A** illustrates the chain graph with microbial genotype predictors, **B** illustrates the chain graph with host exposure predictors, and **C** represents the chain graph with both microbial genotype and host exposure predictors. **A** and **B** represent the chain graphs with single-scale predictors, whereas **C** represents the chain graph with multi-scale predictors.

## Results

The data used for this study were collected from 14 cohorts of pigs, longitudinally sampled from birth to harvest between October 2008 and December 2010 in North Carolina, USA. Pigs were reared under either conventional or ABF production systems. In the conventional farms, animals are reared indoors and slaughtered at 6 months of age. Additionally, animals received antimicrobial drugs for therapeutic and growth promotion purposes. These data were collected before the 2017 implementation of the new Veterinary Feed Directive, which eliminated the use of medically important antimicrobial drugs for animal growth promotion. In ABF farms, animals are reared outdoors and slaughtered at approximately 9 months. Animals in ABF cohorts did not receive antimicrobial drugs. During each of the three swine production stages, *i.e.* farrowing, nursery, and finishing, livestock management questionnaires were administered to producers to ascertain antimicrobial use, biocide use, biosecurity practices, and other animal husbandry practices. Fecal and environmental samples were collected five times over the course of the pigs' lifespan. *C. coli* isolated in the samples underwent broth microdilution antimicrobial susceptibility testing (AST) and were whole-genome sequenced. Sequences were screened for antimicrobial resistance genes using AMRFinderPlus [31]. Isolates with genotypic and phenotypic resistance data were included in the analysis, along with the exposure data generated in the questionnaires.

In the chain graphs, the outcome layer contains the antimicrobial susceptibility data - measured as $log_2$-transformed MIC. The MIC values were not interpreted using breakpoints or epidemiologic cut-offs and $log_2$ MIC values were analyzed as continuous variables. The two predictor layers are the resistance genotype and host exposure layers. The lists and explanations of all included predictor variables (host exposure data and genotypic resistance data) for the conventional and ABF chain graphs are given in S1 Data and S1 Table. The learned chain graph yields the effect estimation of predictor variables, or risk factors, significantly associated with the outcome layer, represented by $\beta$, while simultaneously accounting for the unconditional associations among outcomes, represented by $\Omega$ (Fig 1).

The density of $\beta$ and $\Omega$ edges, and Bayesian Information Criterion (BIC) for each chain graph are given in Table 1. We investigated a range of $\lambda$ and $\rho$ penalties ranging from 0.10 to 0.5. Briefly, these penalties are used to induce sparsity in these models by reducing the smallest coefficients to zero using a soft-thresholding method; higher penalties will produce more sparse and parsimonious models and lower penalties will result in denser and more complex models. In the current analysis, different penalty values were evaluated using model stability, model interpretability, and the Bayesian information criterion (BIC). The chain graphs presented in the main text had a penalty of 0.25 for both $\lambda$ and $\rho$. The density of all $\beta$ edges in the genotype predictor chain graphs were higher than in chain graphs with host exposure predictors. Despite the unequal numbers of predictors considered in conventional versus ABF chain graphs, edge densities of the $\beta$ components were comparable among conventional versus ABF chain graphs for the chain graphs with single-scale predictors, *i.e.* genotype or host exposure predictors. The density of $\Omega$ edges was relatively high across all chain graphs for both production systems (Table 1). Considerable variation in outcomes was not accounted for by the predictor variables evaluated.

### Risk factors for fluoroquinolone resistance

Overall, chain graphs predicting fluoroquinolone phenotypic antimicrobial susceptibility contain few predictors (Figs 2 and 3). Genotype predictors explain most of the variance of ciprofloxacin MIC. In conventional systems, the *gyrA* T86I point mutation alone explained 79% of the variance in ciprofloxacin MIC (Table 2). Its presence corresponded to a 64-fold

**Table 1. Global graphical metrics for all six learned chain graphs in this study. Edge densities represent the proportion of all possible edges that are present in the graph. The densities and number of $\beta$ and $\Omega$ edges in each graph are shown here, along with BICs for each graph.**

| System | Predictors (N) | Density $\beta$ edges | Density $\Omega$ edges | BIC |
|---|---|---|---|---|
| Conventional | Genotypes (11) | 0.26 | 0.61 | 18.43 |
| Conventional | Host exposures (35) | 0.16 | 0.69 | 16.22 |
| Conventional | Genotypes and host exposures (46) | 0.10 | 0.61 | 16.19 |
| ABF | Genotypes (13) | 0.26 | 0.72 | 18.21 |
| ABF | Host exposures (20) | 0.14 | 0.69 | 14.81 |
| ABF | Genotypes and host exposures (33) | 0.16 | 0.72 | 13.79 |

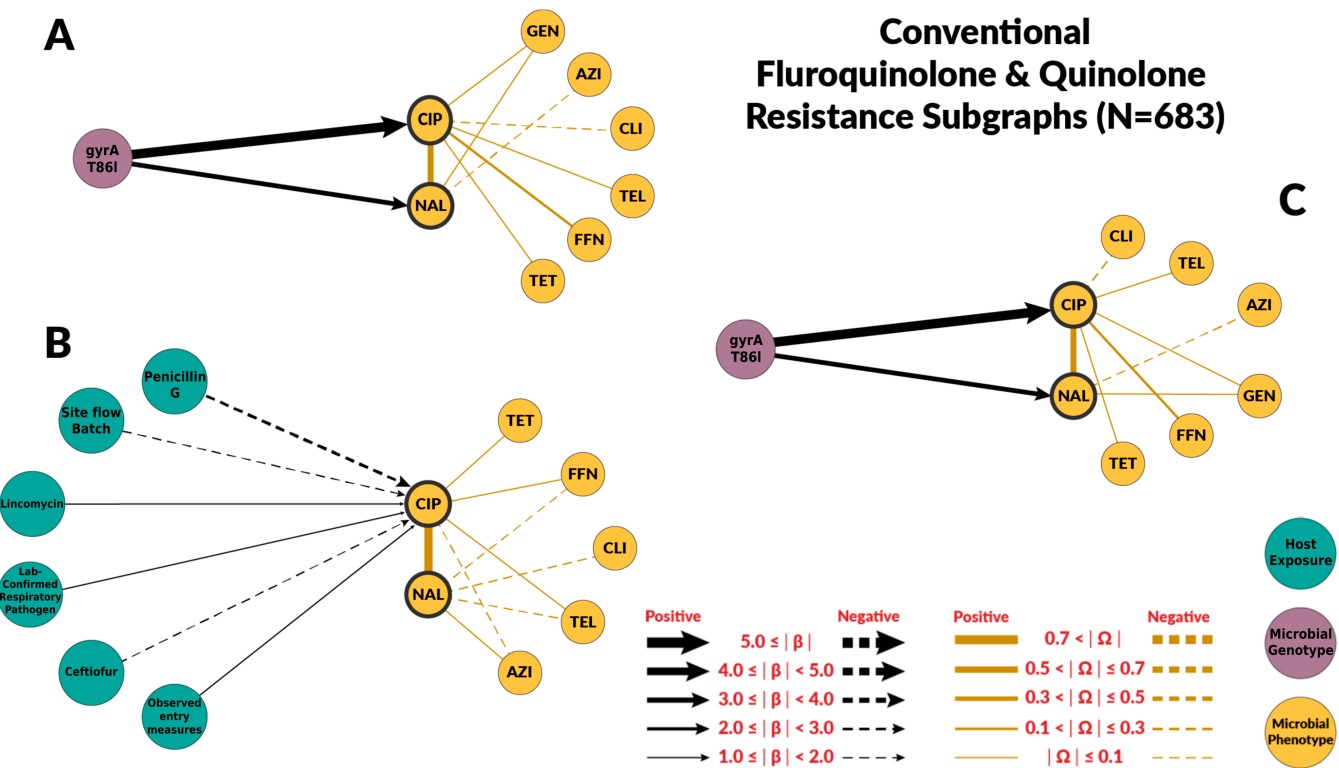

**Fig 2. Chain subgraph focusing on fluoroquinolone and quinolone antimicrobial susceptibility outcomes among *C. coli* from conventional swine farms.** The subgraph consists of all the respective predictor variables adjacent to NAL and CIP, the directed edges between those nodes and NAL and CIP, all of the antimicrobial susceptibility variables, and all the undirected edges between the antimicrobial susceptibility variables. **A** illustrates the subgraph for the genotype single-scale predictor chain graph. **B** depicts the host exposure single-scale predictor chain subgraph. **C** depicts the multi-scale chain subgraph with both genotype and host exposure predictor variables. Host exposure variables are illustrated with teal colored nodes, microbial resistance genotypes as purple nodes, and microbial antimicrobial susceptibility (minimum inhibitory concentrations (MIC)) as yellow nodes. Only predictors with a $\beta$ coefficient equal or greater than 1 are depicted. MIC abbreviations are AZI for azithromycin, CIP for ciprofloxacin, CLI for clindamycin, ERY for erythromycin, FFN for florfenicol, GEN for gentamicin, NAL for nalidixic acid, TEL for telithromycin and TET for tetracycline. Host exposures are Penicillin G (Swine hosts were administered procaine penicilin-G antibiotics (parenteral)), Site flow batch (Batch management of swine hosts at the level of the farm site), Lincomycin (Swine hosts were administered lincomycin antibiotics (parenteral)), Lab-confirmed respiratory pathogen (Infection of a respiratory pathogen on farm confirmed by diagnostic lab), Ceftiofur (Swine hosts were administered ceftiofur antibiotics (parenteral)), Observed entry measures (Biosecurity measures were observed prior to entering the farm).

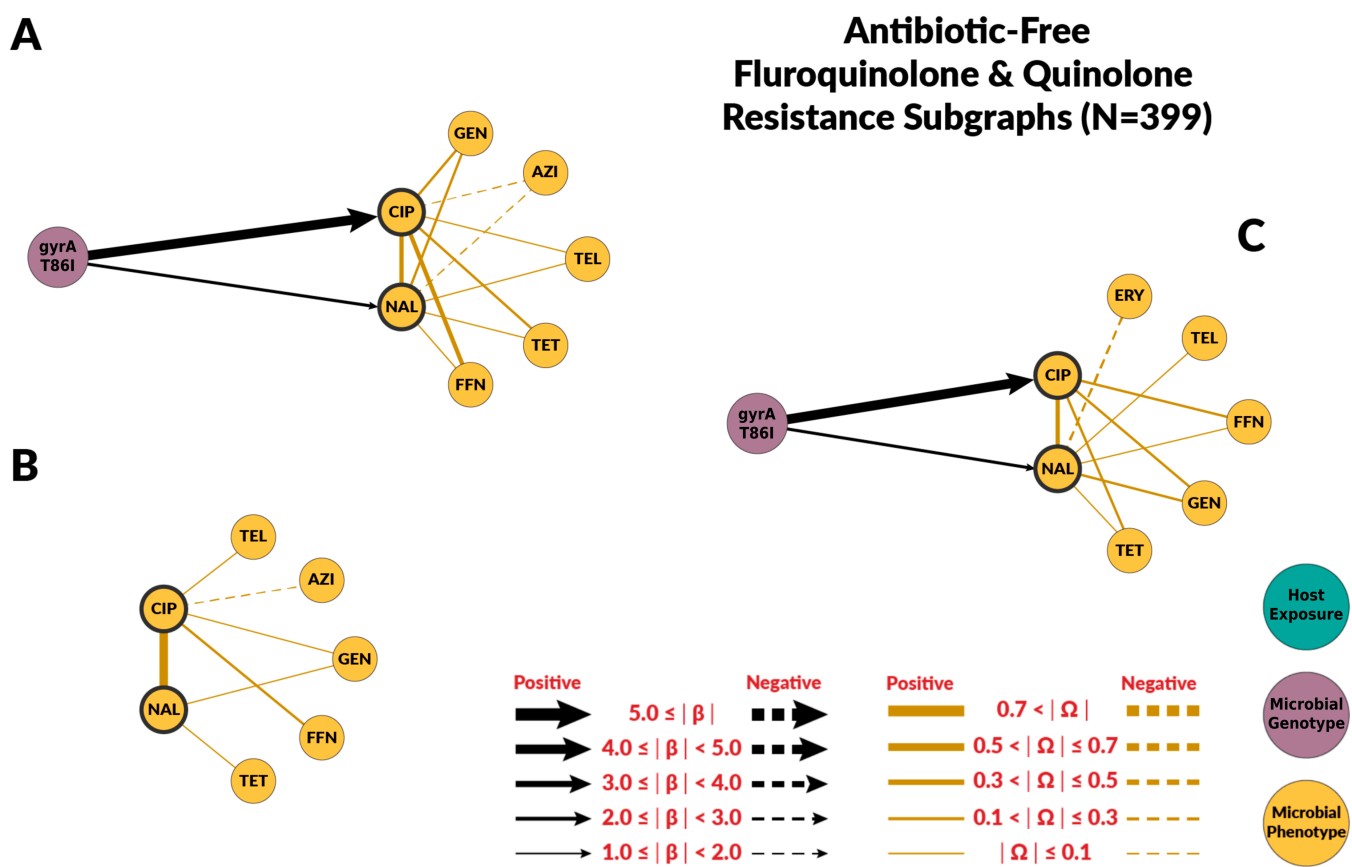

**Fig 3. Chain subgraph focusing on fluoroquinolone and quinolone antimicrobial susceptibility outcomes among *C. coli* from ABF swine farms.** The subgraph consists of all the respective predictor variables adjacent to NAL and CIP, the directed edges between those nodes and NAL and CIP, all of the antimicrobial susceptibility variables, and all the undirected edges between the antimicrobial susceptibility variables. **A** illustrates the subgraph for the genotype single-scale predictor chain graph. **B** depicts the host exposure single-scale predictor chain subgraph. **C** depicts the multi-scale chain subgraph with both genotype and host exposure predictor variables. Host exposure variables are illustrated with teal colored nodes, microbial resistance genotypes as purple nodes, and microbial antimicrobial susceptibility as yellow nodes. Only predictors with a $\beta$ coefficient equal or greater than 1 are depicted. MIC abbreviations are AZI for azithromycin, CIP for ciprofloxacin, CLI for clindamycin, ERY for erythromycin, FFN for florfenicol, GEN for gentamicin, NAL for nalidixic acid, TEL for telithromycin and TET for tetracycline.

dilution increase in ciprofloxacin MIC in both conventional and ABF systems. Similarly, the *gyrA* T86I point mutation was also associated with an increase in MIC for the quinolone drug nalidixic acid. Overall, host exposures were poor predictors of fluoroquinolone resistance. In conventional systems, host exposures alone explained only 35% of the variation in ciprofloxacin MIC (Table 2), and had no predictive value in ABF systems. For the conventional system, the host exposures with the highest coefficients were whether biosecurity entry measures were observed ($\beta_{Obsentrymeas \rightarrow CIP;Conv} = 2.11$), laboratory confirmed respiratory pathogens ($\beta_{labconfres \rightarrow CIP;Conv} = 1.98$), and use of non-fluoroquinolones antibiotics (($\beta_{PenG \rightarrow CIP;Conv} = -2.46$ and $\beta_{ceftiofur \rightarrow CIP;Conv} = -1.89$). No host exposure variables were selected when both genetic and host exposures were considered in both systems (Figs 2C and 3C).

**Table 2. Summary of the variance explained by the predictor layers ($R_x^2$) and the remaining partial correlations ($R_y^2$) for selected microbial resistant phenotypes. The complete list of independent variables and dependent included in each models and the summary for other microbial resistant phenotypes are listed in S1 Table.**

| Microbial phenotype | System | Predictor layers | $R_x^2$ | $R_y^2$ | $R_{total}^2$ |
|---|---|---|---|---|---|
| Ciprofloxacin | Conventional | Genotypes | 79.7 | 14.2 | 94.0 |
| | | Host exposures | 37.5 | 53.5 | 91.0 |
| | | Genotypes and host exposures | 80.3 | 13.8 | 94.2 |
| | ABF | Genotypes | 58.3 | 25.8 | 84.1 |
| | | Host exposures | 6.1 | 68.8 | 74.8 |
| | | Genotypes and host exposures | 60.7 | 23.6 | 84.3 |
| Erythromycin | Conventional | Genotypes | 28.1 | 69.4 | 97.5 |
| | | Host exposures | 15.3 | 82.5 | 97.7 |
| | | Genotypes and host exposures | 34.8 | 62.9 | 97.7 |
| | ABF | Genotypes | 34.3 | 63.4 | 97.7 |
| | | Host exposures | 30.7 | 66.6 | 97.3 |
| | | Genotypes and host exposures | 71.5 | 25.9 | 97.4 |

## Risk factors for macrolide resistance

The chain graphs for the conventional systems explained a relatively low amount of variability observed in azithromycin and erythromycin MICs. The chain graph with both genotypic and host exposures explained only 35% of the variation (Table 2). Conversely, chain graphs with genotypic factors alone or both genotypic and host exposures as predictors explained a higher proportion of the observed variance in the ABF system, up to 76%. The ABF chain graphs were also more dense than the chain graphs for the conventional system.

Among genotypic predictors, the presence of the A2075G point mutation on the *23S rRNA* gene was the most consistent and strongest predictor of macrolide MIC ($\beta_{A2075G \to AZI;Conv}$ = 6.14, $\beta_{A2075G \to AZI;ABF}$ = 7.87, $\beta_{A2075G \to ERY;Conv}$ = 3.96, $\beta_{A2075G \to ERY;ABF}$ = 5.68). In the subgraphs for macrolides, we included the $\beta$ edges from A2075G to clindamycin $\beta_{A2075G \to CLI;Conv}$ = 1.71, $\beta_{A2075G \to CLI;ABF}$ = 3.93) and telithromycin $\beta_{A2075G \to TEL;Conv}$ = 1.71, $\beta_{A2075G \to CLI;ABF}$ = 3.93), as they belong to the drug classes lincosamide and ketolides, respectively. These classes have similar mechanisms of action as macrolide drugs and are often grouped together as the macrolide-lincosamide-streptogramin, or "MLS", family of antibiotics [32]. The A2075G *23S rRNA* mutation was also associated with clindamycin and telithromycin phenotypes, but its effect magnitudes were much smaller (Figs 4 and 5). None of the chain graphs learned in this study found the A103V amino acid substitution of the L22 protein in the 50S ribosomal subunit (*50S L22* A103V) to be associated with MIC to any macrolide or related drug. In addition to A2075G *23S rRNA* mutation, other resistance genes and mutations were associated with macrolides MIC. However, the strength and direction of the association were not consistent across systems or chain graphs (Figs 4 and 5). For example, *blaOXA*-193 gene and *blaOXA*-489 were positively associated with macrolide MIC in conventional farms, but negatively associated in ABF.

Host exposures associated with macrolide resistance were specific to the production system. In conventional farms, the presence of ruminants ($\beta_{Ruminants \to AZI;Conv}$ = 1.31) and observed respiratory signs in the sampled pigs ($\beta_{Obsclinsigns \to AZI;Conv}$ = 2.10) were retained in the chain graph with both genotypic and host exposure factors. For ABF farms, a higher

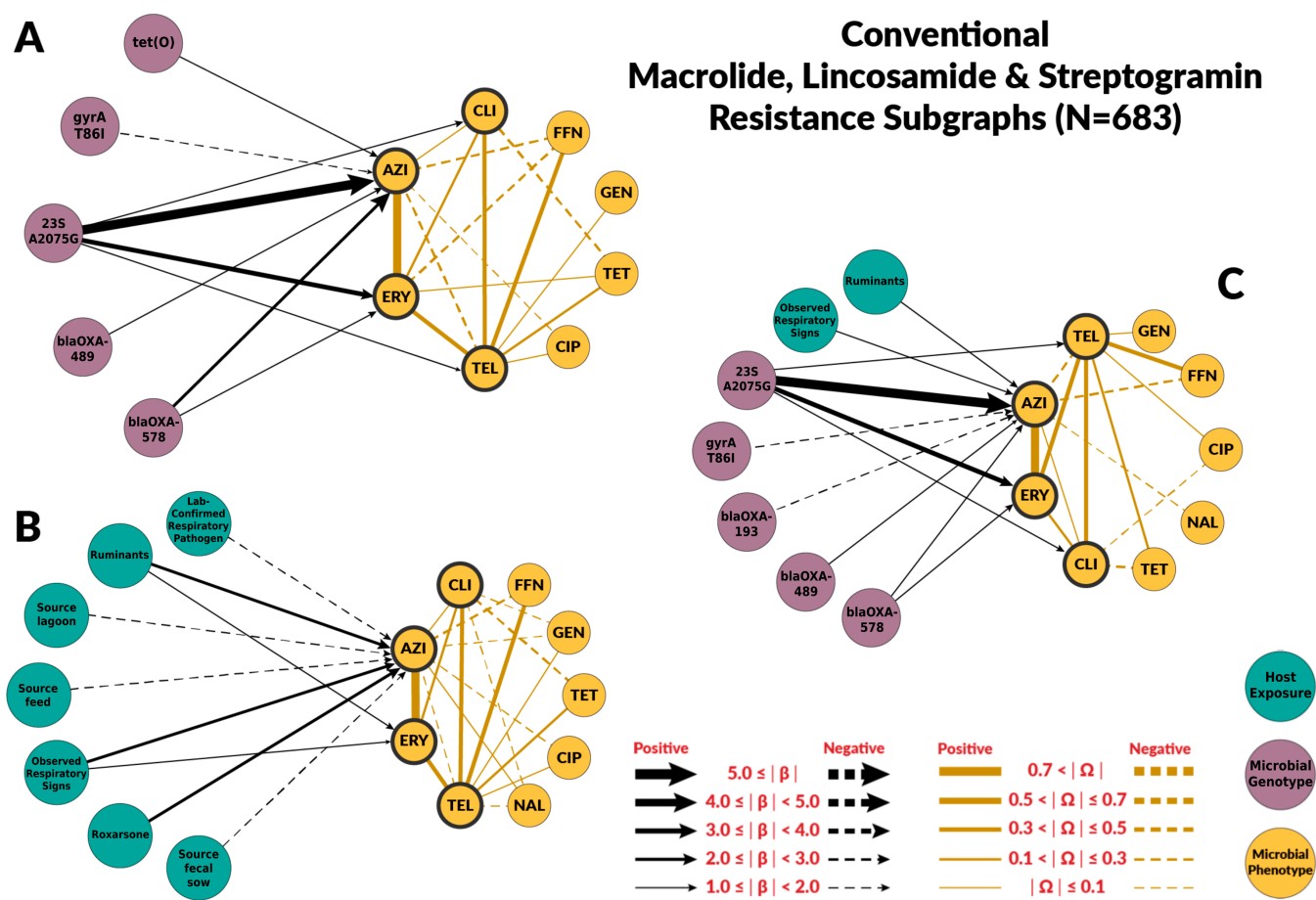

**Fig 4. Chain subgraph focusing on macrolide, lincosamide and streptogramin antimicrobial susceptibility outcomes among *C. coli* from conventional swine farms.** The subgraphs consists of all the respective predictor variables adjacent to AZI and ERY, the directed edges between predictor variables and AZI, ERY, TEL, and CLI, all of the antimicrobial susceptibility variables, and all the undirected edges between the antimicrobial susceptibility variables. **A** illustrates the subgraph for the genotype single-scale predictor chain graph. **B** depicts the host exposure single-scale predictor chain subgraph. **C** depicts the multi-scale chain subgraph with both genotype and host exposure predictor variables. Host exposure variables are illustrated with teal colored nodes, microbial resistance genotypes as purple nodes, and microbial antimicrobial susceptibility as yellow nodes. Only predictors with a $\beta$ coefficient equal or greater than 1 are depicted. MIC abbreviations are AZI for azithromycin, CIP for ciprofloxacin, CLI for clindamycin, ERY for erythromycin, FFN for florfenicol, GEN for gentamicin, NAL for nalidixic acid, TEL for telithromycin and TET for tetracycline. Host exposures are Lab-confirmed respiratory pathogen (Infection of a respiratory pathogen on farm confirmed by diagnostic lab), Ruminants (Presence of cattle, goats, or sheep on farm), Source lagoon (Isolated from lagoon sample), Source feed (Isolated from feed sample), Observed respiratory signs (Clinical respiratory signs were observed among sampled pigs), Roxarsone (Roxarsone administered to pigs), Source fecal sow (Isolated from sow fecal sample).

number of host exposures were retained in the combined chain graph. The host exposures with the higher coefficients were zinc supplementation ($\beta_{Zincsup \to AZI;ABF}$ = 4.64, whether feed was sourced at the farm ($\beta_{Feedsourcefarm \to AZI;ABF}$ = 4.34, and copper supplementation $\beta_{Coppersup \to AZI;ABF}$ = 1.83. In conventional farms, roxarsone, which is also a supplement with heavy metals, was also identified as a risk factor on the host exposure chain graph, $\beta_{Roxarsone \to AZI;Conv}$ = 2.40. S4 Table summarizes the values for all the estimated coefficients on the chain graphs.

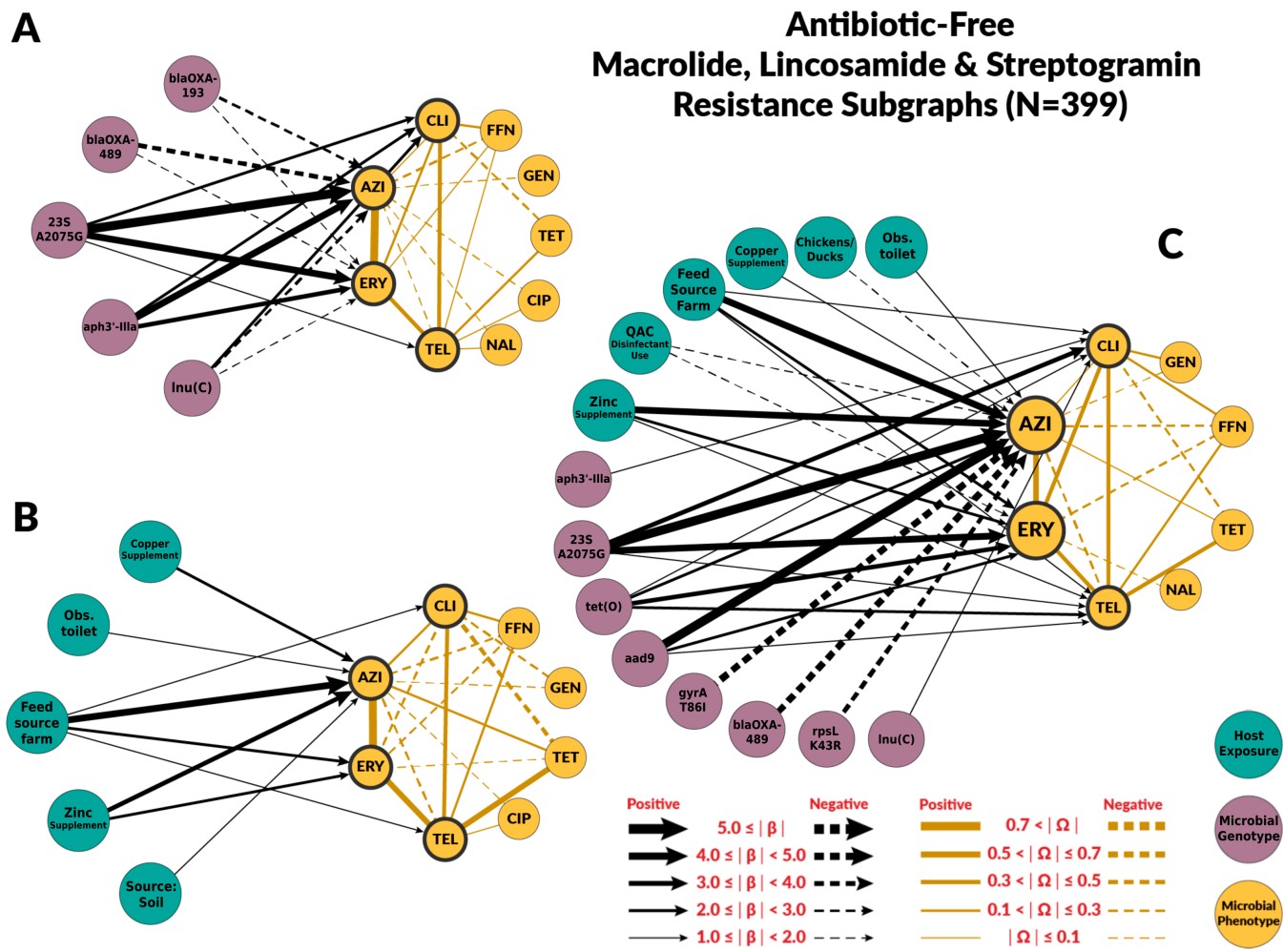

**Fig 5. Chain subgraph focusing on macrolide, lincosamide and streptogramin antimicrobial susceptibility outcomes among *C. coli* from ABF swine farms. A** illustrates the subgraph for the genotype single-scale predictor chain graph. The subgraphs consists of all the respective predictor variables adjacent to AZI and ERY, the directed edges between predictor variables and AZI, ERY, TEL, and CLI, all of the antimicrobial susceptibility variables, and all the undirected edges between the antimicrobial susceptibility variables. **B** depicts the host exposure single-scale predictor chain subgraph. **C** depicts the multi-scale chain subgraph with both genotype and host exposure predictor variables. Host exposure variables are illustrated with teal colored nodes, microbial resistance genotypes as purple nodes, and microbial antimicrobial susceptibility as yellow nodes. Only predictors with a $\beta$ coefficient equal or greater than 1 are depicted. MIC abbreviations are AZI for azithromycin, CIP for ciprofloxacin, CLI for clindamycin, ERY for erythromycin, FFN for florfenicol, GEN for gentamicin, NAL for nalidixic acid, TEL for telithromycin and TET for tetracycline. Host exposures are Copper supplement (Administration of copper supplements), Obs. toilet (Toilet was observed on site), Feed source farm (Feed was sourced on the farm), Zinc supplement (Administration of zinc supplements), Source: soil (Isolated from soil sample), Chickens/ducks (Presence of chickens and/or ducks on farm), QAC disinfectant use (Use of disinfectants containing quaternary ammonium compounds).

## Discussion

Epidemiological analytical studies are broadly concerned with understanding the relationship between an outcome and a set of potentially causal predictor variables (e.g., exposures). Antimicrobial susceptibility outcomes at the isolate level are often multivariate, as MICs for a panel of antibiotics are commonly performed. In epidemiological studies, these are usually simplified by focusing on one resistance outcome at a time and by dichotomizing MICs using epidemiological cut-offs or clinical breakpoints. Both simplifications come with a loss

of information [33]. Chain graphs are a multivariate method that allows us to simultaneously model the joint probability distributions of all antimicrobial susceptibility outcomes and their predictors [22]. This feature is critical to fully investigate multidrug resistance phenomena because high correlations among MICs are common as numerous processes result in the co-selection of resistances, including genetic linkage of resistance genes on mobile genetic elements, cross-resistance among related drugs, and epistatic interactions between genes [34–36].

Our chain graphs showed that resistant genes and mutations were the most important predictors of MICs changes. The largest effects among all chain graphs learned in this study were associated resistance point mutations *gyrA* T86I and *23S rRNA* A2075G as risk factors for fluoroquinolone- and macrolide-resistance in *C. coli*, respectively. These results aligned with previous experimental work indicating that the T86I point mutation in the *gyrA* gene is the most important fluoroquinolone-resistance genotype among *Campylobacter* spp. [37], and that the A2075G *23S rRNA* mutation confers high-levels of macrolide resistance and it is the most predominant macrolide-resistance conferring genotype in *Campylobacter* spp. [38]. None of the chain graphs learned in this study found the A103V amino acid substitution of the L22 protein in the 50S ribosomal subunit (*50S L22* A103V) to be associated with MIC to any macrolide or related drug (Figs 4 and 5). Experimental studies have found that mutations in the L22 ribosomal protein conferred modest levels of macrolide resistance in some bacterial species [39]. As a result, the *50S L22* A103V has been suspected to potentially confer macrolide resistance in *Campylobacter* spp. [38]. However, other experimental studies have not identified similar effects in *Campylobacter* spp. [40], nor have observational studies identified associations between *50S L22* A103V and phenotypic macrolide resistance [41]. Our findings are consistent with these latter studies in *Campylobacter* spp., suggesting the *50S L22* A103V mutation has minimal effect on macrolide resistance in this population of *C. coli*.

The host exposures include risk factors related to biosecurity and antimicrobial, metal, and disinfectant exposures. These exposures generally had less explanatory power than genotype features ($R^2 \leq 35\%$ for all the graphs with host exposures alone). For fluoroquinolones, host exposures were not selected for the combined graph. Some host exposures remained in the combined graph for macrolides, but they had less explanatory power than genotype features. Given the high levels of macrolide resistance in both conventional and ABF farms, we originally hypothesized that antimicrobial exposure would only be identified as a risk factor for fluoroquinolone resistance but not macrolide resistance. For macrolides, no antimicrobial exposure variables were significantly associated with MIC to either macrolide drugs, supporting our hypothesis. We did not find fluoroquinolone exposure to be a risk factor for fluoroquinolone resistance. However, the presence of laboratory-confirmed respiratory pathogens was positively associated with ciprofloxacin MIC. Swine respiratory diseases are the only indication for use of fluoroquinolones in swine [42]. Other antimicrobial drugs were identified as risk factors. Penicillin-G, ceftiofur, and lincomycin exposures were predictors of fluoroquinolone resistance in conventional farms' host exposure chain graph. Ceftiofur and penicillin-G were negatively associated with ciprofloxacin MICs. The most likely explanation is that the exposure to injectable ceftiofur or penicillin-B was a marker for not using enrofloxacin, an injectable fluoroquinolone, as these drugs may have similar indications. Similarly, some of the associations between genes and mutations conferring resistance to antimicrobial drugs outside the macrolide class may be indirect associations arisen from the pattern of antimicrobial drugs use (Fig 4).

Our findings are consistent with other studies that found antimicrobial exposure only partially explained resistance levels in swine farms [43–46]. Despite the obvious role of antimicrobial exposures in selecting for resistance, it is generally difficult to establish epidemiological associations between antimicrobial use and resistance [33]. Antimicrobial use metrics, in our case presence of the exposure, are often a rough measurement of the selective pressures occurring at the individual level. Pharmacokinetics and pharmacodynamics dynamics at the gut level modify the relationship between exposure and resistance levels [47,48]. Additionally, variations on the fitness effects of antimicrobial-resistant genes and mutations complicate the relationship between antimicrobial use and resistance [15,49].

Exposure to heavy metals, including zinc, copper, and arsenic, was found to be associated with higher MICs for azithromycin among *C. coli*. Zinc and copper supplementations were positively associated with macrolide resistance in the ABF farms. Zinc and copper exposure were not included in the conventional chain graphs because all pigs sampled for this study were exposed to zinc and copper. Therefore, conventional pigs had no variation in zinc or copper exposure. However, roxarsone exposure, an organoarsenic compound no longer approved for use in swine, was positively associated with azithromycin's MIC in conventional farms. Co-selection of antimicrobial resistance by heavy metals is a well-documented phenomenon in agriculture settings [50–52]. In other bacterial species, including *C. jejuni*, heavy metals can select for the persistence and expression of multidrug efflux pumps that act on a broad range of antimicrobial drugs, including macrolides [52,53]. Factors related to biosecurity (e.g., presence of other animal species on site or presence of respiratory signs) were related to macrolide antimicrobial susceptibility outcomes. However, the variables and strength varied by farm type and graph. In conventional farms, the presence of ruminants and respiratory signs remain predictors of macrolide antimicrobial susceptibility outcomes in the combined graphs. Ruminants represent other important livestock reservoirs of *Campylobacter* spp. [54,55].

The method presented here represents a new approach to address the complexity of antimicrobial resistance epidemiology. This data-driven approach aims to overcome some of the limitations of more classic analytical approaches. The classic framework relies heavily on testing *a priori* hypotheses informed by existing knowledge that may be unavailable due to the complexity of the processes driving resistance dynamics. These complex data sets also contain many highly correlated variables that may be irrelevant to the research question. Common assumptions from regression analysis, such as linearity or lack of multicollinearity, are often challenging to meet, and the need to run large numbers of hypothesis tests creates further statistical error [56]. Probabilistic graphical models are highly interpretable methods, thus facilitating the interpretation of the identified risk factors and their relation to the modeled outcome [22]. As with other methods in machine learning, however, the identified $\beta$ and $\Omega$ estimates are sensitive to the hyperparameters $\lambda$ and $\rho$ chosen for the chain graph analysis. We used BIC and stability selection methods to identify the $\lambda$ and $\rho$ penalties needed to produce the most computationally stable models of the data [57]. If the selected penalties are too high, then important and informative connections may be inappropriately pruned from the model, and if penalties are too low, then important edges are difficult to identify and the model is challenging to interpret.

Probabilistic graphical models have extensively used for causal discovery [58]. By simultaneously evaluating the effect of multiple resistance genotypes on multiple antimicrobial susceptibility phenotypes within a single model, we uncovered novel genotype-phenotype relationships in natural *C. coli* populations that have not been described before. For example, *aph(3')-IIIA* and *aad9* genes encode enzymes that confer resistance to aminoglycoside antibiotics, and have not been shown to directly confer macrolide resistance [59]. Given the large

magnitudes of effects for *aph(3')-IIIA* and *aad9* seen in the ABF system, it is worth considering ways in which such aminoglycoside-resistance genes can directly or indirectly be related to macrolide resistance. First, it is theoretically possible for the aminoglycoside resistance proteins encoded by these resistance genes to have off-target metabolic effects of *C. coli* that indirectly influence macrolide resistance. Resistance-conferring enzymes have been found to have effects on microbial metabolism beyond their mechanisms of action on their target antibiotic drugs [60]. While this would be interesting to pursue with further research, there is currently no evidence to suggest this to be the case for the enzymes encoded by *aph(3')-IIIA* and *aad9*. Second, these genes may be genetically linked to another genotype that is more logically related to macrolide resistance (e.g. directly confers resistance). Both *aph(3')-IIIA* and *aad9* have been identified on mobile genetic elements, and consequently were genetically linked to each other and genes conferring resistance to other antibiotic classes [61,62]. Additionally, the effect estimates observed zinc exposure on macrolide MIC among ABF *C. coli* highlight promising areas for future experimental, genomic, and epidemiological research to untangle why so much macrolide resistance was observed without antimicrobial drug exposure. It would be inappropriate to draw conclusions regarding causation from this study alone. The edges of the models presented here are based on observational data from a study originally designed to estimate resistance prevalence at different stages of swine production, and was not designed to demonstrate causal relationships between independent and dependent variables. Additionally, the current chain graph learning algorithm selects models for fit using main effects only and does not consider effect measure modification or confounding, which allows for unidentified bias in the estimates.

Finally, with the chain graph methods used in this study, we could not account for microbial population structure, which can influence resistance dynamics, especially for *Campylobacter* spp. Several strains of *C. coli* exist and have distinct biological features that enable them to survive in specific environments better than others [54]. These strains may survive well because of their non-resistant genotypes, but may still incidentally retain specific resistance genotypes that do not necessarily confer a fitness advantage. In this case, population structure could explain why specific resistance genotypes and phenotypes exist in contexts without strong selective pressures for resistance. We could include sequence type (ST) as exposure measuring population structure. However, in this study, the large number of unique STs identified among C. coli made this approach impractical, while clonal complexes (CC) cannot be used in *C. coli* as these all cluster as CC-828 [15].

## Conclusion

In this study, we have demonstrated how a chain graph-based analytical framework can generate insights in the complex epidemiology of antimicrobial resistance. Our approach characterizes context-specific risk factors and identifies future research areas to better understand the mechanisms driving resistance dynamics. In applying this approach to identify risk factors driving *Campylobacter coli* resistance in isolates originated from pigs in antibiotic-free and conventional farms, we demonstrated how for fluoroquinolones, risk factors were similar across both types of farms, but risk factors for macrolides were different across settings. This method identified the use of heavy metals, specifically zinc and copper, as potential selection pressures that are increasing the prevalence of macrolide resistance in ABF-raised pigs.

## Methods

### Ethics statement

The cohort study was approved by the North Carolina State University Institutional Animal Care and Use Committee (protocol #08-031-A).

### Overview of the data

The data used for this study were collected from 14 cohorts of pigs longitudinally sampled 5 times from birth from October 2008 and December 2010 in North Carolina, USA as part of a previous comparative study of AMR in swine [30] reared under either conventional (9 cohorts) or antibiotic-free (ABF, 5 cohorts) production systems. These two systems differed considerably in several management practices. In conventional systems, pigs received antimicrobial drugs in feed, water, and injectables, and were reared indoors. In ABF systems, pigs did not receive antimicrobial drugs for any reason and were reared outdoors. During each of the three swine production stages, *i.e.* farrowing, nursery, and finishing, swine producers filled questionnaires with closed and open ended questions on eight different themes including farm description, herd inventory, swine management, health status, medications and vaccinations, description of the facilities, feed and water, and biosecurity measures. Additionally, a questionnaire was filled out by a researcher at the sampling site regarding the presence of other animals and biosecurity measures. S1 Table contains the list of host exposures used in the chain graphs. Fecal and environmental samples were collected five times throughout the pigs' production: At farrowing (7–10 days of age), after being moved into nursery pens (4 weeks of age), just before being moved out of nursery pens (7 weeks of age), during finishing (16 weeks of age), and prior to harvest (26 weeks of age and 48 hours before transported for slaughter). Fresh fecal samples from 35 healthy pigs and corresponding environmental samples were collected at each of the five production stages five longitudinal sample. Environmental samples included soil, feed, water, and drag swab samples of the floor and structures. All samples were cultured for *Campylobacter* spp., and 2,898 *C. coli* isolates were identified. Minimum inhibitory concentrations (MIC) were determined via broth microdilution antimicrobial susceptibility testing (AST) (Sensititre and CAMPY Sensititre panel; Trek Diagnostic Systems, Ohio). These plates included the following antibiotics in the given concentrations: azithromycin (AZI, 0.015 - 64 $\mu g/ml$), ciprofloxacin (CIP, 0.015 - 64 $\mu g/ml$), erythromycin (ERY, 0.03 - 64 $\mu g/ml$), gentamicin (GEN, 0.12 - 32 $\mu g/ml$), tetracycline (TET, 0.06 - 64 $\mu g/ml$), florfenicol (FFN, 0.03 - 64 $\mu g/ml$), nalidixic acid (NAL, 4 - 64 $\mu g/ml$), telithromycin (TEL, 0.015 - 8 $\mu g/ml$), and clindamycin (CLI, 0.03 - 16 $\mu g/ml$). The MIC values were not interpreted using cutoff points.

Of the 2,898 *C. coli* isolates phenotypically characterized via AST, 1,466 were whole-genome sequenced using the Illumina MiSeq platform as part of the US FDA GenomeTrakr project [63]. Isolates were not sequenced because their cohort had incomplete survey data, or isolates did not grow. Some isolates were also removed because they had insufficient quality scores. DNA was extracted from *C. coli* isolates based on manufacturer protocols using the Qiagen DNeasy Blood and Tissue Kits. DNA libraries were prepared using Nextera XT DNA Library Preparation Kits, then sequenced using Illumina MiSeq v2 (500 cycles) chemistry kits and flow cells. Once whole-genome sequences were obtained and submitted to NCBI, the associated FASTQ files for *C. coli* isolates used in this study were downloaded from GenBank using the fasterq-dump function from the NCBI SRA Toolkit (v2.10.8) [64], *de novo* assembled using the Shovill (vs 1.0.4) implementation of SPAdes (v3.12) [65], then annotated using Prokka (v1.14.6) [66]. Afterward, genomes were screened for AMR genes and other virulence

factors using AMRFinderPlus (v3.10.5) [31]. Resistance genes with >90% coverage and >60% identity were considered present in the genome. Quality scores for all sequenced genomes were ascertained using Quast (v5.0.2) [67]. To be included in the analysis, the genome size had to be 1.4-2.0 Mbp, which is the expected genome size of *Campylobacter* spp. Additionally, only genomes with ≤ 200 contigs, N50 ≥25 kb, and L50 ≤50 were included in the analysis [15]. Of the 1466 *C. coli* that were whole-genome sequenced, 1282 had sufficient quality scores. The Sequencing Read Archives identifiers for all *C. coli* isolates considered for inclusion in this study are listed in S1 Data.

All sequenced *C. coli* isolates with complete AST data and sufficient genomic quality scores were further considered for inclusion in the chain graph analyses. The algorithm to learn chain graphs from data requires complete observations [26]. Of the 1282 *C. coli* with sufficient quality scores, 1082 isolates had complete antimicrobial exposure data and therefore qualified for final study inclusion. Afterward, other host exposure variables with any missing observations or zero variance were excluded from the analysis. Binary variables with very low variance (< 1% prevalence) were also excluded from the study. Collinearities among variables in the dataset were identified as any pair with correlations ≥ |0.7| based on a correlation matrix of all host exposure and genotype variables. Variables were removed from the analysis until no variable pairs with correlations ≥ |0.7| remained. Variables with high correlations to multiple other variables were removed first, and variables expected to be biologically important were retained where possible. The final dataset of conventional *C. coli* isolates included 683 observations and 46 predictor variables, and the dataset for ABF *C. coli* isolates included 360 observations and 33 predictor variables. A list and detailed explanation of all included predictor variables for the conventional and ABF chain graphs are given in S1 Table. The table of co-occurrence of resistant genes and mutations are given in S2 Table.

## Chain graph models

We aimed to identify risk factors at the host exposure and microbial genotype scales for phenotypic macrolide- and fluoroquinolone-resistance among *C. coli* from swine farms. We accomplished this by learning three distinct chain graphs in which $\log_2$-transformed MIC comprised the outcome (phenotypic) data layer and variables from the following layers comprised the predictor data layer: 1) microbial genotype, 2) host exposure, 3) microbial genotype & host exposure (Fig 1). The $\log_2$-transformed MIC values were treated as a continuous variable on a scale where a one-unit increase represented a two-fold increase in MIC, and MIC values were not interpreted using breakpoints, epidemiological cut-offs, or other rules. A $\beta$ edge in each chain graph represents an association between a predictor variable and a specific $\log_2$-transformed MIC. Thus, a $\beta$ edge for a binary predictor variable with a value of 1.0 can be interpreted as follows: after accounting for all other predictor variables in the model, the presence of this binary predictor was associated with an observed 2-fold dilution increase in MIC for a given antibiotic. On the graphs, we only displayed predictor variables with $\beta$ edge magnitudes ≥1.0 because these edges corresponded to microbiologically detectable changes in MIC. The presence of an $\Omega$ edge between two phenotype nodes represents a non-trivial penalized partial correlation among these two MICs after accounting for all outcome variation explained by the predictor variables and other outcome variables. The presence of an $\Omega$ edge, therefore, suggests that other variables that are not included in the model explained this correlation among outcome variables.

Several host exposure variables, such as antimicrobial exposure and biosecurity practices, were highly correlated with the swine production system. Therefore, the graphs were stratified by the production system. In total, six unique chain graphs were learned for this study: three

learned from *C. coli* isolated from conventional swine production systems, and three learned from ABF production systems.

The structure and parameters of these chain graphs were learned using a penalized maximum likelihood method developed by [26]. The algorithm to learn the chain graphs is an iterative process with the following steps: 1) regress the outcome data layer on the predictor data layer, 2) identify the conditional edges ($\beta$) connecting nodes in these different data layers, and 3) identify the partial correlations ($\Omega$) connecting nodes within the outcome data layer. These steps are repeated until model convergence. The resulting $\beta$ edge adjacency matrix encompasses all effect estimates in which a predictor variable had a significant, conditional association with a resistance phenotype outcome after accounting for any variation explained by other predictor variables in the model. The $\Omega$ edge adjacency matrix represents the non-trivial penalized partial correlations after accounting for conditional associations explained by the $\beta$ edge adjacency matrix.

Lasso and graphical lasso (glasso) L1 regularization methods were used to induce sparsity in the $\beta$ and $\Omega$ edge adjacency matrices, respectively [68,69]. The $\lambda$ and $\rho$ penalties determine which $\beta$ and $\Omega$ edges are included in the final model or reduced to zero, respectively. Using lower penalties leads to the inclusion of more edges. All $\lambda$ and $\rho$ penalties used in our chain graph were evaluated using a stability selection procedure [57]. The final penalties were chosen based on model interpretability.

Bootstrapping was used to generate 95% confidence intervals for all $\beta$ and $\Omega$ effect estimates. For each of the six distinct chain graphs, 200 bootstrapped samples were selected. The chain graphs were learned via the same algorithm applied to the whole dataset, except for the stability selection procedure. Instead of deriving unique $\lambda$ and $\rho$ penalties for each bootstrapping iteration, chain graph models for each bootstrapped sample used the same $\lambda$ and $\rho$ penalties identified from the chain graphs using the entire conventional and ABF datasets. The bootstrap sample proportion was half the available sample for each system (340 isolates for conventional and 198 for ABF). The resulting $\beta$ and $\Omega$ effect estimates from each bootstrapping iteration were used to generate 95% confidence intervals. All analyses for this study were conducted in R (v3.6.3).

In addition to the $\beta$ and $\Omega$ edge parameter values, several global graphical metrics were also quantified. The edge density represents the number of edges learned in the network out of all possible edges, and can be quantified for both $\beta$ and $\Omega$ edge adjacency matrices. Additionally, because $\beta$ edges with magnitudes $\geq 1.0$ are of greater interest, the density of edges with magnitudes $\geq 1.0$ were also calculated for each chain graph. To better interrogate qualitative differences in risk factors for each resistance phenotype of interest, we also calculated the densities of all edges and edges with magnitudes $\geq 1.0$ for subgraphs of each resistance phenotype. To assess whether the model fits the data, the BIC was also calculated for each chain graph. Similar to the Akaike Information Criterion, a lower BIC corresponds to models that fit the data better than a higher BIC. All comparisons were considered significant at $\alpha = 0.05$.

To evaluate the relative contribution of the directed and undirected portions of the chain graphs to explain each phenotypic outcome, $Y_i$, two coefficients of determination, $R_X^2$ and $R_{Y_j}^2$ for the predictors ($X$) and the other resistances ($Y_j$), respectively were estimated (Table 2). The value of $R_X^2$ for each outcome $i$ can be estimated directly using the linear models defined in $\beta$ as: $R_{i,X}^2 = \frac{Y_i - \sum^P \beta_{p,i} X_p}{Var(Y_i)}$, where $P$ is the set of predictors. In contrast, linear coefficients cannot be derived from $\Omega$ and $R_{Y_j}^2$ cannot be calculated directly. Instead, the residuals from the former model, $e_i = Y_i - \sum^P \beta_{p,i} X_p$, were fit to linear models with terms selected by nonzero elements of $\Omega$. The coefficients of determination for these residual linear models were calculated as $\frac{Var(\sum^{J_i} B_{j,i} Y_j)}{Var(e_i)}$, where $J_i$ were the set of variables adjacent to $Y_i$ in the chain graph, and $B_{j,i}$ was

the vector of maximum likelihood estimates for the linear model defined by $J_i$. However $R^2_{e_i,Y_j}$ only describes the portion of variance not explained by $\beta_i X$ and we must multiply $R^2_{e_i,Y_j}$ by $1 - R^2_{i,X}$ to calculate $R^2_{i,Y_j}$. Subsequently, the total proportion of variance in each phenotypic outcome explained by the model, $R^2_{i,total}$, may be calculated as the sum of $R^2_{i,X}$ and $R^2_{i,Y_j}$. The set of equations to determine the coefficients of determinations is the following:

$$e_i = Y_i - \sum^{P} \beta_{p,i} X_p \tag{1}$$

$$R^2_{i,X} = \frac{Var(\sum^{P} \beta_{p,i} X_p)}{Var(Y_i)} = 1 - \frac{Var(e_i)}{Var(Y_i)} \tag{2}$$

$$R^2_{e_i,Y_j} = \frac{Var(\sum^{J_i} B_{j,i} Y_j)}{Var(e_i)} \tag{3}$$

$$R^2_{i,Y_j} = R^2_{e_i,Y_j}(1 - R^2_{i,X}) \tag{4}$$

## Supporting information

**S1 Data. Accession numbers.** Complete list of all sequenced *Campylobacter coli* isolates from BioProject PRJNA293228 that were included in the analysis.
(CSV)

**S2 Data. Input data.** Dataset with the input data necessary to run the chain graphs.
(XLSX)

**S1 Table. Predictor variables.** A list and detailed explanation of all included predictor variables for the chain graphs.
(XLSX)

**S2 Table. Co-occurrence.** Resistance genotype co-occurrence tables generated from conventional swine farms and those from ABF swine farms.
(XLSX)

**S3 Table. MIC summary.** A document with a summary of MIC distributions and isolate counts by cohort.
(DOCX)

**S4 Table. Adjacency lists.** Spreadsheets containing linear model coefficients and partial correlations that define the learned models.
(XLSX)

**S1 Code. Code.** Zip file with the R code necessary to run the chain graphs.
(ZIP)

## Author contributions

**Conceptualization:** Christine A. Wang, William J. Love, Cristina Lanzas.

**Data curation:** Christine A. Wang, Siddhartha Thakur.

**Formal analysis:** Christine A. Wang, William J. Love, Cristina Lanzas.

**Funding acquisition:** Christine A. Wang, Cristina Lanzas.

**Methodology:** William J. Love, Cristina Lanzas.

**Resources:** Siddhartha Thakur.

**Supervision:** Cristina Lanzas.

**Visualization:** Manuel Jara.

**Writing – original draft:** Christine A. Wang, Manuel Jara, Cristina Lanzas.

**Writing – review & editing:** Christine A. Wang, William J. Love, Manuel Jara, Arnoud H. M. van Vliet, Siddhartha Thakur, Cristina Lanzas.

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
