## [Decision Letter · Decision Letter 0]

25 Mar 2025

 PCOMPBIOL-D-25-00063

Risk factors for fluoroquinolone- and macrolide-resistance among swine Campylobacter coli using multi-layered chain graphs

PLOS Computational Biology

Dear Dr. Lanzas,

Thank you for submitting your manuscript to PLOS Computational Biology. After careful consideration, we feel that it has merit but does not fully meet PLOS Computational Biology's publication criteria as it currently stands. Therefore, we invite you to submit a revised version of the manuscript that addresses the points raised during the review process.

Specifically, please consider carefully the comments of the reviewers on the inferences of causality and the adequacy/completeness of the methods.

Please submit your revised manuscript within 60 days May 25 2025 11:59PM. If you will need more time than this to complete your revisions, please reply to this message or contact the journal office at ploscompbiol@plos.org. Please include the following items when submitting your revised manuscript:

We look forward to receiving your revised manuscript.

Kind regards,

Narendra M Dixit

Academic Editor

PLOS Computational Biology

Thomas Leitner

Section Editor

PLOS Computational Biology

**Journal Requirements:**

At this stage, the following Authors/Authors require contributions: Christine Annie Wang, William J Love, Manuel Jara, Siddhartha Thakur, Arnoud H.M. van Vliet, and Cristina Lanzas. Please ensure that the full contributions of each author are acknowledged in the "Add/Edit/Remove Authors" section of our submission form.

4) Please amend your detailed Financial Disclosure statement. This is published with the article. It must therefore be completed in full sentences and contain the exact wording you wish to be published.

1) If the funders had no role in your study, please state: "The funders had no role in study design, data collection and analysis, decision to publish, or preparation of the manuscript."

2) If any authors received a salary from any of your funders, please state which authors and which funders..

**Reviewers' comments:**

Reviewer's Responses to Questions

**Comments to the Authors:**

**Please note that one of the reviews is uploaded as an attachment.**

Reviewer #1: The review is uploaded as an attachment

Reviewer #2: The authors describe a novel approach for determining risk factors for antimicrobial resistance in Campylobacter coli in swine using multi-layered chain graphs. This approach compared to traditional regression models has the advantage of being able to take into account a larger number of predictor and outcome variables. Multi-layered chain graphs were built to determine risk factors in both conventional and antibiotic free swine production systems using genotypic data, environmental/host factor exposure data, and a combination of both genotypic and host factor exposures as predictors. Thank you to the authors for a well-written and interesting manuscript. One major concern is the causal relationship between exposures, genotypes, and phenotypes. Otherwise, there are only minor comments.

Major comments:

The graphs were set up to evaluate the impact of both resistance genes or mutations and host exposures on antimicrobial phenotypes (MICs). However, this ignores the relationship between host exposures and resistance genes/mutations. If, as seen in the fluoroquinolone graphs, mutations drive much of the variation in phenotypes, then the mutation could be considered to be a mediator between host exposure and phenotype (outcome). Within a directed acyclic graph framework, conditioning on a mediator reduces the causal effect between exposure and phenotype, which could explain why no host exposures are significant in the fluoroquinolone graphs when both exposures and genotype are included. The genomic layer should be separated from the exposure layer, and edges from the exposure to genomic layer should be added.

Minor comments

Author summary: Campylobacter coli is not italicized.

Line 10: Campylobacter is not italicized

Introduction:

Lines 66-77: What is the hypothesis for fluoroquinolone resistance? It is mentioned in the discussion, but it would be helpful for readers to state it here along with the macrolide resistance hypothesis.

Methods:

Line 271: How many of the 18 cohorts were from ABF farms vs conventional?

Line 289: How were the isolates selected for sequencing? Randomly? Do they represent all isolates? If the selection is not representative of all the isolates studied, then this should be discussed as a limitation of the study and the bias estimated, if possible.

Line 317: How was one variable in the correlated pair selected for exclusion over the other?

Line 339: In MIC quality control, often one 2-fold dilution is acceptable variation. Therefore, shouldn’t a beta > 2 be the cut-off for a microbiologically detectable change?

It is unclear how the fluoroquinolone and macrolide specific graphs were created as only the general approach is described. For example, for the fluoroquinolone graph, were only the fluoroquinolone phenotypes included in the outcome layer (line 352)? Or were only the fluoroquinolone phenotypes eligible for connection to other layers?

Results:

Line 92: include which software was used to screen sequences for AMR genes. It was detailed in the methods, but since the results come first in this manuscript, it would help the reader.

Line 96: Would it be possible to change “predicted layers” to “predictor layers” to avoid confusion with predictors vs outcome variables?

Line 102: Were β and Ω supposed to be represented in Figure 1?

Line 104: Please include a brief explanation of what λ and ρ are in the results section since the methods description comes later.

Line 105: How was the penalty of 0.25 selected?

The edges between MICs (Ω edges) could represent expected cross-resistance phenotypes of antimicrobial drugs within the same classes (for example, ciprofloxacin and nalidixic acid or azithromycin and erythromycin). Calculating density of these edges within-classes vs between-classes could provide additional insight into differences in resistance relationships across systems, and exposures.

Figure 2/3/4/5: Can the phenotype node layout be the same in each panel for easier comparisons? Could the coefficients be written on each edge?

Lines 136-137: It is unclear what is meant by including the beta edges to clindamycin and telithromycin. β edges are shown in black and connect different graphs, but the wording implies connecting different MICs, which would be the Ω edges. The confusion here relates to the lack of specificity in how the drug-specific graphs were developed in the methods.

Table 2:

• In the caption, please list which supplemental material appendix specifically contains the independent and dependent variables included in the models.

• Is it possible to put the antimicrobial name abbreviations at the bottom of figures to make it easier for the reader rather than searching in the body of the text?

• Why is the variance explained by prediction layers Rx2 for ciprofloxacin ABF genotypes and host exposures together lower than Rx2 for genotypes or host exposures alone?

Please add descriptive statistics on the MIC distributions for each drug and system, similar to Table S2. It is also helpful to include the range of MIC values tested on the Sensititre plate, as this creates censoring of the data.

Discussion:

Line 185: Give the table that the R2≤35% statistic was cited from.

Lines 196-200: Could a similar mechanism be behind the relationship between beta-lactamase, aminoglycoside, and tetracycline resistance genes and macrolide MICs?

Lines 239-241: What is the impact of a smaller or larger penalty on the presented results?

Line 246: Which genotype-phenotype relationships are novel? The resistance genes and mutations studied are known to result in phenotypic resistance or reduced susceptibility.

Conclusion: Please include a conclusion specific to the results, beyond just the novelty of the method.

**Have the authors made all data and (if applicable) computational code underlying the findings in their manuscript fully available?**

Reviewer #1: Yes

Reviewer #2: Yes

PLOS authors have the option to publish the peer review history of their article (what does this mean?). If published, this will include your full peer review and any attached files.

Reviewer #1: **Yes: **Brittany L Morgan Bustamante

Reviewer #2: No

**Figure resubmission:**
---

## [Decision Letter · Decision Letter 1]

10 Jul 2025

PCOMPBIOL-D-25-00063R1

Risk factors for fluoroquinolone- and macrolide-resistance among swine Campylobacter coli using multi-layered chain graphs

PLOS Computational Biology

Dear Dr. Lanzas,

Thank you for submitting your manuscript to PLOS Computational Biology. After careful consideration, we feel that it has merit but does not fully meet PLOS Computational Biology's publication criteria as it currently stands. Therefore, we invite you to submit a revised version of the manuscript that addresses the points raised during the review process.

Please submit your revised manuscript within 30 days Sep 09 2025 11:59PM. If you will need more time than this to complete your revisions, please reply to this message or contact the journal office at ploscompbiol@plos.org. Please include the following items when submitting your revised manuscript:

We look forward to receiving your revised manuscript.

Kind regards,

Narendra M Dixit

Academic Editor

PLOS Computational Biology

Thomas Leitner

Section Editor

PLOS Computational Biology

**Additional Editor Comments:**

Comments of Reviewer 1 must be addressed before publication.

**Reviewers' comments:**

Reviewer's Responses to Questions

**Comments to the Authors:**

Reviewer #1: Thank you for the thoughtful changes and additions to the manuscript based on reviewer comments. I believe the manuscript is much improved and have only a few additional minor comments that need to be addressed:

1. If qualitative interpretations for MICs were not used, then “phenotypic resistance” is not a correct term to use throughout the manuscript. It appears the authors only assessed resistance through genotypic resistance. This needs to be updated throughout the manuscript because it is confusing to see “phenotypic resistance” when qualitative interpretations were not used.

2. Results, line 85: given the topic of antibiotic use in livestock is somewhat contentious and highly scrutinized, I would recommend the authors be very deliberate and careful with their language here. In the US, using medically important antibiotics for growth promotion is illegal (although I recognize the data for this study was collected before that law was enacted). It might be worth updating this to clarify that subtlety–potentially even just saying something about how as this was prior to the U.S. law change making MIADs... or something similar.

3. Results line 133: Is this a typo? Should it say, “The host exposures WITH the higher coefficients”?

4. Methods line 359: Is there a word missing between "the" and "on"?

Reviewer #2: Lines 109-114: please review for typos. “less than penalty”, “evaluated using on”

Lines 132-135: please review for typos

Line 286: please review for typos. “it is worth consider”

Line 359: please review for typos. “The on these plates”

**Have the authors made all data and (if applicable) computational code underlying the findings in their manuscript fully available?**

Reviewer #1: Yes

Reviewer #2: Yes

PLOS authors have the option to publish the peer review history of their article (what does this mean?). If published, this will include your full peer review and any attached files.

Reviewer #1: **Yes: **Brittany Lauren Morgan Bustamante

Reviewer #2: **Yes: **Casey L. Cazer

**Figure resubmission:**
---

## [Editor Report · Decision Letter 2]

24 Jul 2025

Dear Dr. Lanzas,

We are pleased to inform you that your manuscript 'Risk factors for fluoroquinolone- and macrolide-resistance among swine Campylobacter coli using multi-layered chain graphs' has been provisionally accepted for publication in PLOS Computational Biology.

Best regards,

Narendra M Dixit

Academic Editor

PLOS Computational Biology

Thomas Leitner

Section Editor

PLOS Computational Biology

---

## [Editor Report · Acceptance letter]

PCOMPBIOL-D-25-00063R2

Risk factors for fluoroquinolone- and macrolide-resistance among swine  *Campylobacter coli* using multi-layered chain graphs

Dear Dr Lanzas,

I am pleased to inform you that your manuscript has been formally accepted for publication in PLOS Computational Biology. Your manuscript is now with our production department and you will be notified of the publication date in due course.

With kind regards,

Anita Estes
